# The Phenotype-Based Approach Can Solve Cold Cases: The Paradigm of Mosaic Mutations of the *CREBBP* Gene

**DOI:** 10.3390/genes15060654

**Published:** 2024-05-22

**Authors:** Giulia Bruna Marchetti, Donatella Milani, Livia Pisciotta, Laura Pezzoli, Paola Marchisio, Berardo Rinaldi, Maria Iascone

**Affiliations:** 1Università degli Studi di Milano Statale, 20122 Milano, Italy; giulimrc@gmail.com (G.B.M.); livia.pisciotta@asst-fbf-sacco.it (L.P.); paola.marchisio@policlinico.mi.it (P.M.); 2Fondazione IRCCS Ca’ Granda Ospedale Maggiore Policlinico di Milano, 20122 Milano, Italy; 3Child Neuropsychiatry Unit, ASST Fatebenefratelli Sacco, 20100 Milano, Italy; 4Laboratorio di Genetica Medica, ASST Papa Giovanni XXIII, 24127 Bergamo, Italy; lpezzoli@asst-pg23.it (L.P.); miascone@asst-pg23.it (M.I.); 5Medical Genetics Unit, Fondazione IRCCS Ca’ Granda Ospedale Maggiore Policlinico, 20122 Milano, Italy; berardo.rinaldi@policlinico.mi.it

**Keywords:** creb binding protein, Rubinstein–Taybi syndrome, mosaicism, correlations, genotype–phenotype

## Abstract

Rubinstein–Taybi syndrome (RTS) is a rare genetic disorder characterized by intellectual disability, facial dysmorphisms, and enlarged thumbs and halluces. Approximately 55% of RTS cases result from pathogenic variants in the *CREBBP* gene, with an additional 8% linked to the *EP300* gene. Given the close relationship between these two genes and their involvement in epigenomic modulation, RTS is grouped into chromatinopathies. The extensive clinical heterogeneity observed in RTS, coupled with the growing number of disorders involving the epigenetic machinery, poses a challenge to a phenotype-based diagnostic approach for these conditions. Here, we describe the first case of a patient clinically diagnosed with RTS with a *CREBBP* truncating variant in mosaic form. We also review previously described cases of mosaicism in *CREBBP* and apply clinical diagnostic guidelines to these patients, confirming the good specificity of the consensus. Nonetheless, these reports raise questions about the potential underdiagnosis of milder cases of RTS. The application of a targeted phenotype-based approach, coupled with high-depth NGS, may enhance the diagnostic yield of whole-exome sequencing (WES) in mild and mosaic conditions.

## 1. Introduction

Rubinstein–Taybi syndrome (RTS1 MIM 180849; RTS2 MIM 613684) is a rare genetic disorder with an estimated incidence of 1/100,000–125,000. The main clinical features include post-natal growth delay, facial dysmorphisms, broad and duplicated thumbs and halluces, and intellectual disability (ID) [1]. The phenotypic spectrum is notably variable, extending beyond the typical hallmarks to include various congenital anomalies in the central nervous system (CNS), heart, genitourinary system, gastrointestinal tract, and skin [1]. Recently, an international consensus has been released to guide the diagnosis through specific clinical criteria [2]. RTS causative variants affect two highly conserved genes: *CREBBP*, encoding for the cAMP response element binding protein (CREB) binding protein (OMIM *600140, located in 16p13.3) [3], and *EP300*, encoding for the EA1-associated protein p300 (OMIM 602700, located in 22q13) [4]. Both genes play a pivotal role in epigenetic modulation through histone acetylation and chromatin remodelling, meaning that RTS is included within the “chromatinopathies” [5]. Its clinical presentation exhibits wide variability, with no established genotype–phenotype correlation. However, studies suggest that *EP300* pathogenic variants may manifest milder features compared to *CREBBP*, particularly regarding facial appearance and ID [4]. Studies also suggest that variants not involving the histone acetyltransferase (HAT) domain may underlie less severe phenotypes [6]. In addition, somatic mosaicism, although appearing extremely rare, can lead to milder manifestations: only four RTS cases due to mosaic *CREBBP* copy number variants (CNVs) have been previously reported [7,8,9]. These patients showed a less severe phenotype than those with constitutive deletions. Only one previous report described a patient carrying a 3 bp mosaic deletion in *CREBBP*, not extending beyond the gene [10]. Here, we present the first case of a mosaic *CREBBP* single-nucleotide variant (SNV) resulting in a very mild RTS phenotype, and we review the previously described cases.

## 2. Case Report

The patient was the firstborn child of a healthy non-consanguineous couple with an unremarkable family history. Prenatal ultrasound revealed a femur length in the lower centiles. Childbirth was induced at 41+3 gestational weeks; she weighed 2945 gr, her length was 46.5 cm, her occipitofrontal circumference (OFC) was 33.5 cm, and her Apgar score was 9/10 [11]. As a newborn, a sacral dimple was noticed, without spine involvement. Her growth has always been reported to be in the lower centiles and a mild developmental delay was noticed: she first walked independently at 18 months and showed speech impairment (at 2 years old, she could pronounce few words). At the age of 2, she was diagnosed with Duane’s syndrome [12] and suffered from recurrent urinary tract infections. Abdominal ultrasound showed a moderate ectasia of the pyelocaliceal cavities. Both cardiological (EKG and echocardiogram) and neurological (EEG, brain MRI) assessments performed at the age of 2 years old were within normal limits. A first neuropsychiatric evaluation at the age of 4 identified mixed specific developmental disorder and childhood emotional disorder, with a Children’s Global Assessment Scale score < 50.

At our last physical evaluation at the age of 5, her weight was 17 kg (10–25th centiles), height was 105 cm (10–25th centiles), and OFC was 48.6 cm (3rd centile). Facial dysmorphisms included medially sparse and slightly arched eyebrows, broad nasal bridge with a convex nose, prominent columella, high arched palate, and retrognathia. Broad thumbs and halluces and foetal finger pads were noticed bilaterally. 

An updated neuropsychiatric evaluation through WPPSI III performed at the age of 7 years excluded the presence of cognitive delay (QIT: 98), with a lower score in processing speed (QVP: 76).

## 3. Materials and Methods

Once informed consent was obtained, DNA was extracted from leukocytes using the standard procedure, and trio whole-exome sequencing (WES) was performed as described before [13]. Briefly, the exonic and flanking splice junctions’ regions of the genome were captured using the Clinical Research Exome v.2 kit (Agilent Technologies, Santa Clara, CA, USA), and sequencing was performed on a NovaSeq6000 Illumina system with 150 bp paired-end reads. Reads were aligned to human genome build GRCh37/UCSC hg19 and analysed for sequence variants using a custom-developed analysis tool [13]. Additional sequencing technology and variant interpretation protocols have been previously described [13]. On the de novo variants, a 20% allele frequency filter was applied.

A DNA sample from buccal mucosa was collected with the Oragene• DNA (OG-575; DNAGenotek, Ottawa, ON, Canada) kit and isolated according to the manufacturer’s instructions. Subsequently, the *CREBBP* variant’s flanking region was amplified by standard PCR with specific primers (forward: AAGAATGTGGGCTTCTGGTG, reverse: ATACACCCCAAACACGAAGG), fragmented and adapter-ligated with standard protocols (Illumina Nextera, San Diego, CA, USA), and sequenced on a MiSeq Nano flowcell with a very high depth. 

## 4. Results

The “first read” Trio-WES, requested without a specific suspicion but only reporting the patient’s clinical features, returned a negative result for pathogenic variants related to phenotype. After clinical re-evaluation, considering our strong suspicion for RTS, a visual inspection of the *CREBBP* and *EP300* genes allowed the identification of a heterozygous truncating variant in *CREBBP* (NM_004380.3: c.2012C>A, p.(Ser671Ter)) (Figure 1A). The variant was shown in 11/114 reads, with an estimated 20% mosaicism rate. As expected, both parents tested negative for this variant, which was absent from international databases, including Clinvar (https://www.ncbi.nlm.nih.gov/clinvar, accessed on 6 January 2023), LOVD v.3.0 Build 28 (https://www.lovd.nl/, accessed on 6 January 2023), and GnomAD (https://gnomad.broadinstitute.org/, accessed on 6 January 2023). Consequently, the variant was classified as probably damaging according to ACMG guidelines [14]. To confirm mosaicism, a second analysis was performed on DNA extracted from the buccal swab: the variant was found in almost 20% of the reads (coverage 3747) (Figure 1B). The different mosaicism percentages observed between tissues are presumably related to the timing of mutation onset and to sample cell composition.

## 5. Discussion

Today, the wide availability of genetic tests continuously extends our understanding of genetic disorders, but it also carries the risk of a limited use of a phenotype-based approach [15]. Conversely, clinical geneticists can play a key role in unravelling elusive diagnoses and atypical presentations. In this scenario, mild phenotypes, such as those in mosaic patients, pose a huge challenge which threatens to lead to missed diagnoses. Nuanced RTS cases such as *EP300*, not HAT domain or mosaic-mutated patients, may remain undiagnosed using a genome-wide approach. Moreover, if diagnostic handles are lacking and clinical presentation is not typical, phenotypically discerning the right diagnosis can be difficult even for highly experienced clinical geneticists. Mosaicism in RTS, particularly for SNVs, is a rare and intriguing phenomenon that has not been extensively explored nor clinically described. Prior mosaic reports for RTS have predominantly focused on CNVs [7,8], with only one documented case of a mosaic 3 bp intragenic deletion in *CREBBP* identified solely in the buccal mucosa [10]. This patient presented a severe ID with speech delay, short stature, and some RTS-typical dysmorphic features such as teeth talon cusps and broad thumbs and halluces. Less common RTS features reported in the aforementioned case include coloboma and bilateral syndactyly of the second to fourth digits of hands and feet. One further case of somatic *CREBBP* exon 1 duplication, found only upon buccal swab, was reported by Gucev et al. [9]. Here, we report the first case of a mosaic *CREBBP* SNV in a patient with RTS. Our patient’s subtle yet distinctive features, such as facial dysmorphisms and slightly enlarged thumbs, played a pivotal role in guiding the diagnosis. Nonetheless, looking back at these cases, our patient and patient 66 reported by Bentivegna [7] scored a likely RTS diagnosis (see Table 1), proving the recently released diagnostic guidelines have good specificity [2]. Notably, gestaltic hallmarks of RTS, such as low-hanging columella, a convex nasal bridge, and broad thumbs or halluces, which confer a high score to the aforementioned patients, are questionable features that are not univocally recognizable. Moreover, in mosaic RTS patients, other key features of this condition, such as growth defects and neurodevelopmental disorders, may be absent or very minor (see Table 1). Finally, not all cases of *CREBBP* mosaicism can be detected, even using this score (see Table 1). These data reinforce the need for an experienced clinician and a close collaboration with the laboratory to reach a diagnosis in nuanced presentations. 

Notably, despite the absence of cardinal features of RTS such as an intellectual disability, the case presented was nonetheless exhibiting suggestive phenotypic signs that led to further genetic investigations.

Our patient, together with previous mosaic reports, corroborates the strong dosage sensitivity of *CREBBP*: despite the low mosaicism rate, mutations in this gene result in a mild but still recognizable (to an expert eye) phenotype (see Table 1). According to these reports [7,8,9,10], patients harboring mosaic CNVs of *CREBBP* demonstrate a typical RTS phenotype, not always distinguishable from that of patients carrying an SNV. Intriguingly, also for Cornelia de Lange syndrome (CdL), another chromathinopathy with well-documented mosaicism phenomena, studies do not agree on whether patients with a mosaic status display phenotypical features as severe as those with constitutive pathogenic variants [16,17]. Post-zygotic mosaicism is an established phenomenon in various chromatinopathies. Among others, mosaicism for *NIPBL* mutations is a well-documented and not uncommon event in CdL with an estimated rate of as high as 13.1% [18]. The mosaicism phenomena observed in CdL might be related to a genetic reversion of a germinal pathogenic variant that, reducing the fitness of mutated cells, undergoes selective somatic rescue [18]. This paradigm of negative selection may extend beyond CdL and elucidate mosaicisms observed in other chromatinopathies, including RTS. In alignment with current recommendations for CdL [18], we believe that individuals exhibiting a suggestive RTS phenotype, negative for pathogenic variants in peripheral blood, should undergo further investigations, eventually using DNA isolated from different tissues. Also, a more in-depth analysis can have important implications in recurrence risk definition. In this regard, we should consider that parental mosaicism may be undetectable using only blood to check inheritance, and that cases of mosaic RTS could remain undiagnosed [19,20,21,22,23,24].

Moving forward, the development of a specific RTS episignature, as proposed by Aref-Eshghi et al. [25], could serve as a valuable tool to identify cases requiring more in-depth investigations. The episignature, derived from a combination of clinical and genetic features, holds the potential to refine diagnostic strategies and enhance the accuracy in identifying mosaic conditions.

Moreover, according to the recent release of specific management guidelines, achieving an RTS diagnosis allows access to a tailored follow-up [2]. 

Moving to a gene-based point of view, our case enriches the *CREBBP*-related spectrum, which, in addition to the widely variable RTS phenotype, also includes two other conditions with distinctly different clinical features: Menke–Hennekham syndrome 1 (OMIM #618332), linked to pathogenic variants in exons 30 and 31 [26] and the Chromosome 16p13.3 duplication Syndrome [27].

In conclusion, this case report not only advances our knowledge on mosaic variants in RTS but also highlights the intricate interplay between clinical expertise and genetic technologies in arriving at a precise diagnosis. The identification of a mosaic *CREBBP* SNV in our patient prompts a reconsideration of diagnostic protocols, highlighting the need for tight collaborations between clinical geneticists and laboratories to navigate the complexity of chromatinopathies and ensure accurate diagnoses for individuals with rare genetic disorders.

## Figures and Tables

**Figure 1 genes-15-00654-f001:**
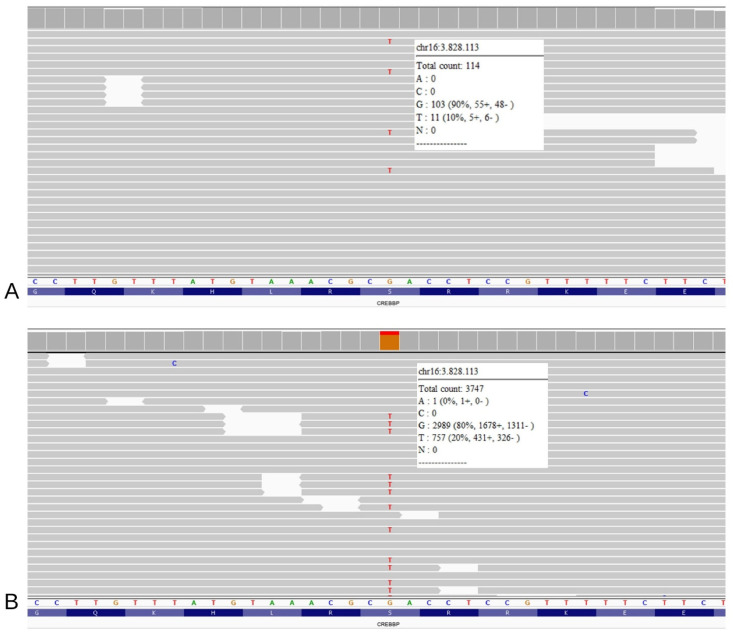
NGS analysis on the patient’s DNA extracted from the blood sample (**A**) and from the buccal brush sample (**B**) reveals the presence of the truncating variant (GRCh37: g.3828113G>T, c.2012C>A, p.(Ser671Ter)). In both panels, the single reads are randomly ranked by the software; refer to the white squares for the total coverage and percentage of wild type reads versus mutated reads.

**Table 1 genes-15-00654-t001:** Clinical features of our patient and mosaic *CREBBP* cases previously reported in the literature and their diagnostic score according to recently defined diagnostic criteria [2].

	**Present Case**	**Patient 38** **[7,8]**	**Patient 40** **[7,8]**	**Patient 66** **[7,8]**	**Gucev et al., 2016** **[9]**	**De Vries et al., 2016** **[10]**
Sex	F	M	F	M	F	M
Age	5	5	26	6	9	11
Facial features: if 3/6 criteria are met, assign 3 points; 4 points if d and/or f is positive						
a. Highly arched eyebrows	+	−	−	−	+	+
b. Downslanted palpebral fissures	−	−	−	+	+	−
c. Convex nasal ridge	+	+	−	−	+	+
d. Columella below alae nasi	+	+	−	+	+	−
e. Highly arched palate	+	−	−	−	−	−
f. Typical smile	−	−	+	−	+	+
Delayed development and/or intellectual disability (2 points)						
	+	+ severe	+ severe	+ severe	+ mild	+ severe
Skeletal: 3 points if b and/or c is positive, or 4 points if a (with or without b/c) is positive						
a. Angulated thumbs and/or halluces	−	−	−	−	+	−
b. Broad thumbs	+	−	+	+	+	+
c. Broad halluces	+	+	+	+	+	+
Growth: 2 points if a and/or b is positive						
a. Microcephaly	+	−	+	−	ND	+
b. Postnatal growth retardation	+	−	+	−	+	+
Supportive: 1 point if c is positive or 3 points if a and/or b (with or without c) is positive						
a. Maternal pre-eclampsia	−	−	−	−	−	−
b. Keloids	−	−	−	−	−	−
c. Hypertrichosis	−	+	−	−	−	+
Diagnostic criteria score	11	6	7	9	12	12
**RTS diagnosis**	**Likely**	**Possible**	**Possible**	**Likely**	**Definitive**	**Definitive**
*CREBBP* mosaic alteration	c.2012C>A, p.(Ser671Ter)	Deletion 3′	Deletion 5′	Deletion 5′	Exon 1 duplication	c.5039_5041delCCT, p.(Ser1680del)

## Data Availability

Data are contained within the article.

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
