# Peer review of "The Phenotype-Based Approach Can Solve Cold Cases: The Paradigm of Mosaic Mutations of the CREBBP Gene"

_genes, 2024, doi:10.3390/genes15060654_

Round 1

Reviewer 1 Report

Comments and Suggestions for Authors

In this case report, Marchetti and colleagues describe a female individual with Rubinstein–Taybi syndrome caused by a mosaic mutation in the CREBBP gene. They describe the phenotype of the affected individual and outline how combined phenotypic and genetic screening were employed for diagnosis. They discuss this case in the context of a limited review of the field and discuss considerations for future diagnostic criteria. The article is well written and clear and provides important insights regarding Rubinstein–Taybi syndrome specifically, and the identification/diagnosis of heterogeneous chromatinopathies more generally. I recommend publication if the following minor comments are addressed.

Title

line 3

‘of CREBBP Gene’ should be - of the CREBBP gene (note there are a number of similar minor errors throughout)

Abstract

line 23

‘Enlisted’ is the wrong word to use in this context

1.Introduction

line 37

‘incidence of 1/100-125,000’ should be - incidence of 1/100,000-125,000

2.Case Reports

General – this section goes through a series of phenotypic characteristics in chronological order up to the age of 7 then concludes with a physical summary at the age of 5. It would be helpful to ensure that all characterisations are documented in the appropriate order providing explicit aging throughout. For example, when was Duane syndrome recognised and when were the EKG and EEGs performed?

line 64

‘OFC’ and ‘Apgar’ – abbreviations should be outlined in full at first use or if describing a key clinical metric, a reference to their meaning should be given.

line 66

‘she walked’ should be – she first walked

Line 72

When introducing Duane syndrome a reference is required.

Line 74-75

‘Within limits’ should read within normal limits

3. Materials and Methods

Line 86

‘was done on’ should read was performed on

Lines 91-94.

From the description it is unclear why it was negative to start with and then re-inspection of the data allowed you to find the variant. Was this a limitation in the algorithm applied? It would be helpful to clarify further what is meant here.

Line 95

You claim ~10% mosaicism due to variant frequency – however given that there are two alleles and this is presumably heterozygous is this not suggestive of 20% mosaicism?

Lines 103-104

‘region have been’ should read region was

Line 104

‘Specific primers’. Primer sequences should be provided as a useful resource.

Line 106

The variant frequency was identified as being 20% in buccal samples. Firstly, there appear to be far more than double the frequency of G>T transitions in panel B so is this true? If it is true and this is simply a sampling issue in the reads shown then at least there should be a comment about why you think the Percentages are different between blood and buccal samples (e.g. cell composition / origin of mosaicisms / technical reasons).

Figure 1. Resolution is too low for details to be usefully discernible. Also, a schematic including the gene locus, the architecture of the gene (simple domain structure etc – HAT domain etc) and the location of the primers would be useful.

4. Discussion

Lines 123 / 124 / 152

Abbreviations CNV / SNV / Cdl all should be given in full at first use. In the case of Cdls you do give this but it comes after first use on line 156-157.

Other minor

All numbers and units throughout should be presented with a space – eg. 5 bp not 5bp or 5-bp.  

Author Response

Thank you very much for taking the time to review this manuscript. In agreement with your annotations, we have modified the manuscript (corrections in red in the re-submitted file). Please find the detailed responses below.

Apart from minor changes, we have revised the case description providing detailed age reference throughout. 

As you suggested, we included references for the Apgar score and Duane's syndrome.

Finally, concerning minor comments, we only point out that CNV and SNV were fully outlined in line 56 and 60. 

Regarding your comment about the analysis, thank you for you precious suggestions and comments, following them we ameliorated the main text.

Below we further discuss on main issues:

  • We included PCR primers' sequences in the updated manuscript
  • First read WES trio negativity based on your input we clarify this in the text: from a clinical point of view we didn't request a targeted analysis but a wide one, just reporting patient's clinical features. Also, for de novo variants, the Laboratory applies a filter on variant frequency at 20%.
  • Mosaicism rate is definetely 20%, thank you for your correction
  • The different mosaicism percentage observed between tissues is presumably related to the timing of mutation onset and to sample cell composition

Regarding the figure: we know it is a low definition one but unfortunately we can only capture it while using the analysis software. Therefore we were not able to acquire a higher resolution one. We are afraid of furtherly loose definition creating a composed image but if you believe it would greatly improve the work we will definitely work on it. 

Reviewer 2 Report

Comments and Suggestions for Authors

Generally, the paper is interesing and the finding worth showing to the medical community.

Some issues:

The figure is low quality /low resolution. 

There is no Results section. The Results section should containd the data on the genetic finding as well as analysis of published papers and variant databases (rather than in discussion).

I do not agree with the view that we should diagnose "milder cases" as RTS. I have patients with likely pathogenic CREBBP variants and no RTS. There are published papers with no-RTS in patients with CREBBP variants. This should be mentioned. There seem to be more phenotypes associated with CREBBP variants.

The table with patients with :mosaic" RTS would be helpful.

Comments on the Quality of English Language

No major issues.

Author Response

Thank you very much for taking the time to review this manuscript. In agreement with your annotations, we have modified the manuscript (corrections in red in the re-submitted file). Please find the detailed responses below.

Regarding the figure: we know it is a low definition one but unfortunately we can only capture it while using the analysis software. Therefore we were not able to acquire a higher resolution one. 

We included a "Results section" and mentioned no "RTS phenotypes" in the discussion. 

A table detailing CREBBP mosaic patients phenotypes was included in the original manuscript, we do not find it in the revised version of the text and believe that the editors may intend to insert it as Supplementary material. You can find it in the revised version of the manuscript attached here. 

Round 2

Reviewer 2 Report

Comments and Suggestions for Authors

The manuscript is satisfactory  now. 

Comments on the Quality of English Language

Minor editing needed.